# Models and Methods of Designing Data-Centric Microservice Architectures of Digital Enterprises

**Sergey Deryabin *** , **Igor Temkin, Ulvi Rzazade and Egor Kondratev**

Institute of Information Technology and Computer Science, Department of Automated Control Systems, National University of Science and Technology MISiS, Moscow 119049, Russia
* Correspondence: deryabin.sa@misis.ru

**Abstract:** The article is devoted to methods and models of designing systems for the digital transformation of industrial enterprises within the framework of the Industry 4.0 concept. The purpose of this work is to formalize a new notation for graphical modeling of the architecture of complex large-scale systems with data-centric microservice architectures and to present a variant of the reference model of such an architecture for creating an autonomously functioning industrial enterprise. The paper provides a list and justification for the use of functional components of a data-centric microservice architecture based on the analysis of modern approaches to building systems and the authors' own results obtained during the implementation of a number of projects. The problems of using traditional graphical modeling notations to represent a data-centric microservice architecture are considered. Examples of designing a model of such an architecture for a mining enterprise are given.

**Keywords:** DEAL 1.0; DEA 1.0; digital enterprise architecture; data-centric microservices architecture; digital transformation of enterprises; Industry 4.0; graphical modeling language; software design; autonomous production

## 1. Introduction

Today, one of the main trends in the development of technological production is the digital transformation of enterprises within the framework of the Industry 4.0 concept [1–6] (Figure 1). The term digital transformation is understood as a transition to a qualitatively new level of business process implementation, which implies minimizing or completely excluding (where possible) human participation in technological work [1,2,7–9]. It means that currently implemented business processes with direct human participation, including with the help of information and automated systems, should be reduced to an autonomous executable software and hardware form. Thus, the key vector of digital transformation of industrial enterprises is the development of new software and hardware intelligent systems based on modern and promising technologies and tools of Industry 4.0 [1–11].

To one degree or another, most industrial enterprises are now actively working in the field of development, implementation, and pilot testing of robotic complexes with various levels of autonomy, including unmanned aerial vehicles, unmanned transport systems and technological installations, as well as the use of certain artificial intelligence methods for solving problems of monitoring, planning, and managing technological processes [6,8,9,12,13]. A relatively new direction can be attributed to the concept of a Digital Twin of an enterprise, which is a high-precision dynamic virtual representation of the enterprise, which has a two-way control connection with its physical counterpart and represents the de facto finalizing part of the digital transformation [2,11–23].

At the same time, despite success in the practical implementation of individual technologies, the issues of organizing effective integration of all solutions in the form of a complete autonomous production system remain obvious and critical problems on the way to their scaling and, in fact, to the actual digital transformation of enterprises [1,3–5,8,14,15,17].

First of all, this problem is related to the need to make significant changes in the structural and functional schemes for implementing business processes of enterprises, which involve reworking the operated hardware and software systems to a data-centric microservice type of architecture. The complexity of processing operational systems is due both to the need for additional economic investments in already operating software products, and to the high risks of disrupting the continuity and safety of production processes (including the associated economic costs) when putting "raw" systems into trial operation.

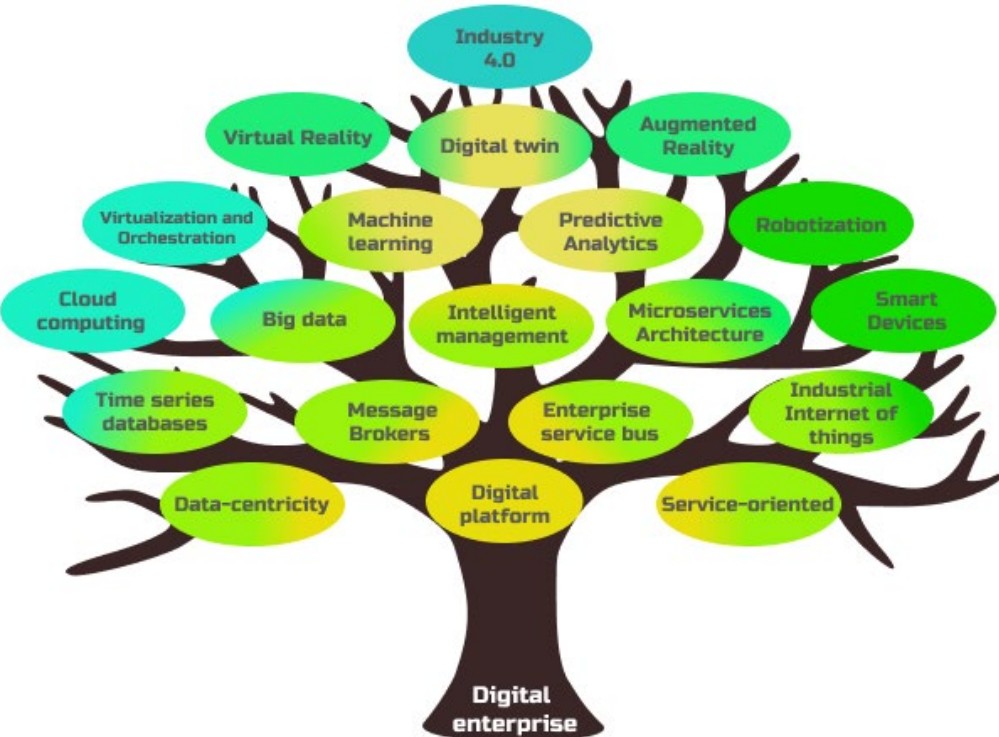

**Figure 1.** Industry 4.0 elements included in the digital enterprise architecture.

The second part of the problem is the lack of a general understanding of the model of such an architecture and, in particular, some generalized unified representation of its components and their structural and functional relationships. Thus, in order to implement such large-scale projects as the digital transformation of industrial enterprises, the issues of developing competent graphical models of the future structural and functional architectures of an integrated digital enterprise management system play significant roles due to the peculiarities of the formulation of initial requirements [24–27], namely:

- The need to find a compromise when compiling and reading the model between users who have competencies in understanding business processes and do not understand the software implementation of systems, and developers who do not have knowledge of the subtleties of business processes, but require a clear understanding of the tasks assigned to them [21,28–30];
- The complexity of formalization, and in some cases a complete lack of understanding of the final functional and non-functional requirements for a digital enterprise as a single system [3–7,19,24–26];
- The need to take into account the existing enterprise architecture, which, among other things, may include the execution of individual business processes in "manual mode" in order to bring it to microservice and data-centric types [30–35];
- The inclusion in the architecture of the future system of many loosely coupled new and promising technologies that have not yet been implemented at the enterprise at the time of the start of design, as well as the lack of successful examples of successful integration of such solutions [36–43].

Existing methodological approaches to the development of models of system architecture, such as, for example, "4 + 1" [44] and "C4" [45], which offer a specification of the minimum and sufficient set of diagrams that need to be formed at the design stage for subsequent development of system, in our opinion, do not fully meet all the requirements set out. Thus, the principle of hierarchical communication of elements inherited in all methodological approaches from the upper "conceptual level" of the architecture to the lower "physical deployment level" is suitable only for small or new systems, since the diagrams used inevitably become unreadable due to the redundancy of structural elements and relationships between them [27,29,30,34,37,44,45]. In addition, only at the conceptual level, the process of drawing up an architecture model may be completely incomprehensible to people who do not have specific skills in system design and knowledge in the field of notation, and, at the same time, the participation of industry specialists in the process of drawing up an architecture model is absolutely necessary, since this guarantees the correctness of the model and, consequently, minimizing the number of errors in its subsequent software implementation. Based on the above, we can conclude that there is a need for a design methodology that considers the main requirements of digital transformation of enterprises and that has the properties of expressiveness and compactness.

Thus, the key objective of this study is to form proposals to improve the efficiency of the design processes of modern complex large-scale systems based on the generalization of approaches to the presentation of a data-centric microservice model of the architecture of an autonomously functioning production system for the "digital" (i.e., a digitally transformed) enterprise using original approaches to graphical modeling that adequately meet the new requirements of Industry 4.0 and traditional architectural design methodologies, as well as the formal presentation of an example model of such an architecture.

## 2. Formalizing a Data-Centric Microservice Architecture

### 2.1. Microservices

Let us look at some aspects of Industry 4.0 technologies in the framework of digital transformation of enterprises to determine generalized requirements for architecture design. As already noted, one of the key features of the digital enterprise architecture is its implementation in a microservice form [30,32–35]. A microservice is an elementary (indivisible) functional task from the list of implemented business processes, which can (and should) be performed by software in offline mode, or (unavoidable in the case of a phased digital transformation of enterprises) in automated or manual mode by an enterprise specialist. At the same time, the main indicator of the actual digital transformation of an enterprise is the transition from manual or automated task execution mode to their autonomous execution by software microservices.

The process of interaction of microservices in the implementation of business processes of an enterprise should be regulated using specialized infrastructure software components, united by the general term "Digital Platform" [25,31,32,37]. The main features of the platform part of the microservice architecture are the following:

- Use of a common integration bus with a flexible application program interface [9,11,16,24,25,27,32,34–38];
- Formalization of the relationship between microservices (information exchange process) in the terminology of "publisher–subscriber" and regularization of such relationships by the message broker [45–48].

The implementation of these components in the architecture of a digital enterprise is determined by the need to provide primary data received from end nodes to all interested parties. At the same time, the microservices themselves are such end nodes, and the publisher–subscriber relationship extends virtually to each of the nodes; i.e., each microservice acts both as a data publisher and as a subscriber. The functions of message brokers are formulated as managing such relationships based on the principle of defining and declaring a topic/topic on the integration bus from the publisher, connecting all subscribers

who need topic data to this topic, and directly converting data to the desired form before transferring it from one component to another.

There are a number of advantages of such an information interaction scheme due to the availability of data for any components, but the microservice paradigm does not imply the stationarity of the architecture itself in terms of the composition of its elements. This means that at some point in time, certain components can be excluded, replaced, or added to the overall architecture, and therefore, there is a dynamic in the technical plan of the entire system's functioning. At the design stage, however, formalizing such dynamics of structural relationships between functional components is not a trivial task. In addition, it is worth noting that the very conditions of information exchange between microservices in such a formulation are a dynamic process with non-stationary parameters of components (data relevance, accuracy, format, etc.), as a result of which the broker must have some aspects of "intelligence", representing a set of message analysis functions. In other words, the task of a broker in the architecture of an autonomously functioning system is not only to compare supply and demand for data (creating a pipeline) but also to control the contents of such connections—by manipulating the position of message packets in the queue. However, at the moment we do not know specific examples of the implementation of such functions, and their formalization is a subject for a separate scientific work.

Moreover, speaking about the non-stationary component composition of the architecture, we mean that the work of functional microservices that ensure the implementation of business tasks of the enterprise should be controlled by a number of service components. By such components we mean the following: quality control services that track performance indicators of functional microservices, such as, for example, performance (calculation time) and accuracy of results. Expressed in the terminology of the agent approach and understanding an autonomously functioning production system as a rational agent, these components act as "critics". Other components are microservice management services—orchestration and virtualization, performing the functions of enabling/disabling or redistributing the placement of microservices in case of their poor quality work. At the same time, disabling one microservice assumes, if at all possible, its replacement by another component performing the same specified function. The connection of a replacement component is possible by accessing external resources—services of software vendors or specialized services, such as source code repository storage systems. However, this approach is currently difficult to implement, carries significant risks in the field of information security and, as a result, requires separate careful consideration.

### 2.2. Data-Centricity

The concept of data is also being reinterpreted in the context of digital transformation to the form of "data as a service", and today there is such a concept as "data-centric architecture" [25,27,45–52]. Data-centricity means building the entire architecture of interaction between microservices and the logic of their work around data. This approach assumes that the system architecture should consider the specifics of how functional components work with data in order to implement their autonomous operation. In other words, microservices must have knowledge of their own functionality, defined as data structures that are sent to external inputs and outputs. However, this approach is not obvious at all and is practically not interpreted at the initial stage of architecture design. We do not deny the critical importance of the data themselves in the system being designed, but still, the system design and development process should initially rely on the goals and objectives of business processes and not on the signs of their implementation (i.e., the appearance of data in general).

Regarding the functional purpose of data as a service, it is assumed that they should describe sufficiently all the parameters of the enterprise state to allow autonomous execution of business processes by functional services. In this formulation, the issue of organizing primary data becomes essential, the volumes of which will create a high load on the industrial network and, of course, limit the time of their processing on the integration bus until

microservices receive it. In order to reduce the load on the network and increase the speed of working with data, today it is proposed to change the format of their representation from primary nodes (smart and robotic devices) to layouts in the form of special objects—time-series data packages (time-series data object) [37,46,48,51]. Such an object is organized according to the principle of uniformity and uniformity of the data contained in it over a certain time interval. At the same time, regarding the new format of data representation as a time series object, the databases of industrial systems that form the enterprise's distributed information storage should be reorganized accordingly; i.e., existing relational databases should be subject to significant changes in the internal organization of data storage by casting them in the form of time series databases.

At the same time, the method of storing data in distributed storage should be regulated even at the stage of forming data packages by publishers before they get on the integration bus. This means that in such tasks as, for example, the organization of dispatching control of autonomous robotic agents of a transport system, it makes sense to create a certain buffer gateway that provides information interaction of agents within the technological environment, and data transfer from agents to the common integration bus of the enterprise is carried out only after the layout of homogeneous data (by type of agents and structure of the data itself) in accordance with certain schemas. This approach can be explained by the fact that one of the primary subscribers to the topic "technical agent telemetry" is a dispatching service that performs the functions of centralized management of technical agents. Accordingly, for such a service, it does not matter whether it receives data for each of the agents separately or receives data from all homogeneous agents in a single packet, assuming that data are received simultaneously in both cases. Moreover, for the service components—the message broker, the integration bus, and the database management system—it would be much more convenient and easier to receive data in one package, thereby reducing the time for the processes of receiving, storing, and providing data.

Therefore, it can be determined that data in a digital enterprise needs to be stored, processed, and transmitted in the form of a specific object—a time series. However, functional microservices with the same purpose, i.e., their internal computational models, can use different parameters of a time series (row size, data frequency, data set in general, etc.). As a result, it is also quite difficult to determine at the design stage the unified parameters (scheme) of the layout of primary data from homogeneous agents in one package for transmission to a common integration bus. It is necessary either to take into account the specifics of each specific existing functional microservice of the same type from different manufacturers, which, of course, is very laborious, or to have some conceptual apparatus of their general principles of operation (i.e., to build on known common functions) and tools for interpreting data packet processing for them.

The second way to solve the problem of working with data can be divided into two components:

- Functional microservices must independently declare to the platform's service modules the data layout schemes that they need to perform their tasks; i.e., in fact, they must have some knowledge of their own abilities. This eliminates the need for unification of data layout schemes for the same type of systems from different manufacturers;
- During the design stage, however, the specifics of working with data should be considered; i.e., graphical interpretations are needed that illustrate the schemes of microservices working with data at the level of abstraction of key data modification methods for their transformation by a broker on the integration bus.

### 2.3. Conceptual Model of the Digital Enterprise Reference Architecture

It is worth noting that most enterprises today do not have all the described properties of a data-centric microservice architecture, which is necessary for a full-fledged digital transformation [1–7]. Some of the information, automation, and other systems involved in the implementation of individual business processes, although maybe having elements of a service-oriented paradigm, still require significant improvements.

As a result, the integration of Industry 4.0 technologies aimed at the digital transformation of the enterprise can (and should) be carried out primarily by working out the techno-working project documentation of the future system and gradually making changes to the methods of organizing and functioning its own business processes. To do this, we suggest the following considerations to be guided by:

1. Redesign of the structural and functional architecture of business process implementation to a microservice type, with formalization of the order of interaction of microservices to the type of publisher–subscriber relations;
2. Organization of autonomous receipt of primary data to all interested parties, carried out by formalizing the functionality of microservices and providing a formal representation of the required data structures to ensure their operation.

In particular, it should be noted that each individual enterprise is a unique case of a set and settings of components of the structural and functional architecture of business process implementation, which seriously complicates the process of formalizing unified requirements for the architecture of a digital enterprise.

Thus, with enterprises of the same industry and similar in terms of organization methods and technologies used, sometimes even located in the same holding company, often the same business processes can be carried out in one case manually by specialists, and in another case using information and automated or autonomous systems from completely different manufacturers and, as a result, with a different configuration. As a result, it is almost impossible to rely on specific architecture examples to formulate general requirements for a digital enterprise.

Based on the understanding of very common properties, i.e., directly executed business processes, such as, for example, dispatching, production planning, and enterprise resource management, the following structure can be determined:

1. Conditional division of the entire architecture into a technological environment in which various technical agents operate (smart devices, piloted, robotic and unmanned equipment), and an information management or distributed computing environment, which is a symbiotic poorly delimited mixture of information, automated or other classes of software systems.
2. Identification of key information and control contours that directly affect the technological environment dispatcher (operational) management systems, specialized technological systems, and enterprise resource management systems.
3. Various requirements (and actual possibilities for their implementation) for the reliability and performance of technical, software, and information support for technological and information management environments.
4. The presence of rigid client-server vertical links both between the agents of the technological environment and the contours of information and management systems, and between the information and management systems themselves, in which the processes of information interaction for the implementation of business processes are carried out using proprietary software interfaces.
5. Availability of similar lists of information to describe the entire problem environment: historical, current, and forecast states of technical and infrastructure agents; historical, current, and forecast planned production indicators, as well as sets of control actions to achieve such indicators; and sets of elementary (indivisible) functional tasks of implemented business processes, indicators of quality metrics of their execution, as well as their inherent lists of data received at the input and produced at the output as a result of work.

All the listed generalized features of modern industrial enterprises for the implementation of digital transformation should be taken into account or, speaking, for example, about point 4, completely eliminated. Therefore, the main guideline for ensuring autonomous execution of business processes is the need to increase interoperability between agents of the technological environment and information management systems by bringing the

schemes of their organization and interaction to the data-centric microservice type of architecture [9,10,15,17,20–27,30–42,45–47]. By such architecture, we mean the following:

1.  Availability of unified application programming interfaces to ensure information interaction processes based on a single and flexible communication template.
2.  Regularization of information interaction processes using specialized service software components—integration buses, brokers, and message queues.
3.  Tracking performance indicators and managing individual agents and systems according to their functional characteristics (accuracy) and technical implementation (performance), by including components such as computing resource orchestration and virtualization services in the architecture.
4.  Organization of receiving primary data of the "lower" (executive) level to all interested parties with minimizing the load on the data transmission environment by reducing messages to a specific form—time series objects, using appropriate time series databases, as well as specialized service components for data modification—mappers.
5.  Division of all business processes into elementary (indivisible) tasks—microservices that have knowledge about their own functional needs and capabilities (incoming and outgoing data) and are organized as "black boxes" in relation to each other; i.e., they work independently of the operation and configuration of other microservices.

A generalized example of a conceptual model of the reference architecture of a digital enterprise is shown in Figure 2.

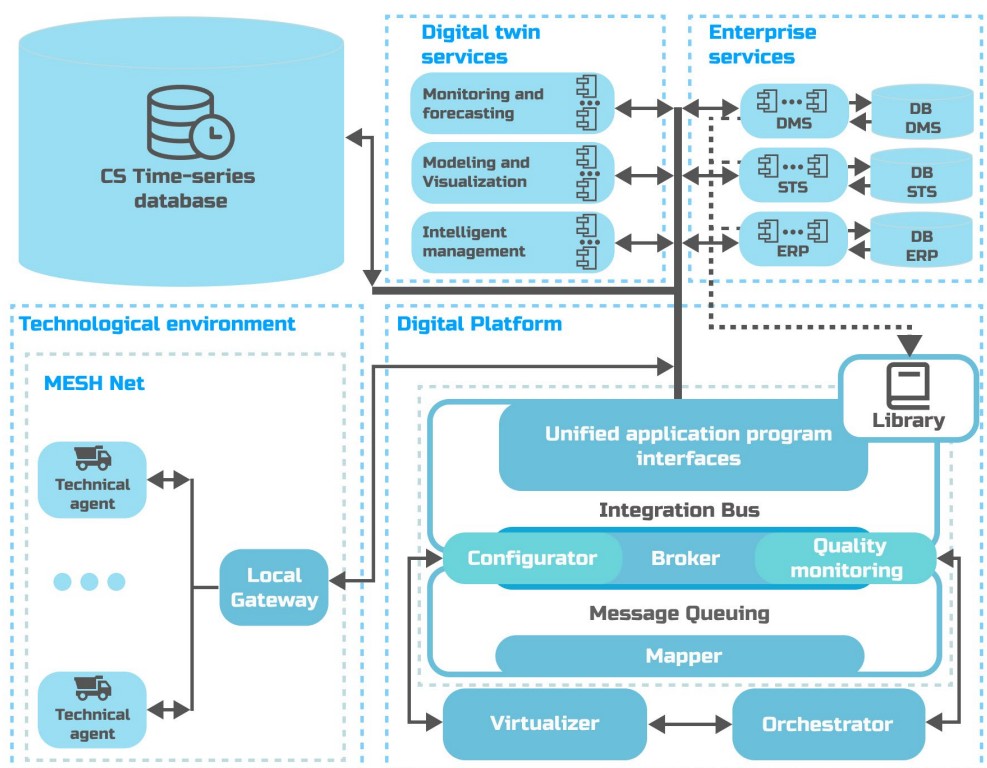

**Figure 2.** Industry 4.0 elements included in the digital enterprise architecture: CS—control system, DMS—dispatcher management system, STS—specialized technological systems, ERP—enterprise resource management systems, DB—database.

However, as noted earlier, it is impossible to implement all the listed components of the data-centric microservice architecture without making significant changes to the systems used at enterprises. As a result, to implement the digital transformation of the enterprise, it is necessary to implement the following steps:

1. Ensure the integration landscape by working out the project techno-working documentation to bring the enterprise architecture in line with the requirements set out, as well as roadmaps reflecting the list of necessary changes.
2. Develop and implement the basic platform part of the architecture, which includes all the necessary service components involved in the organization of information exchange between agents and information and management systems.
3. Redesign or replace the operated systems with components with microservice architecture, i.e., divide business processes into separate independent elementary functional blocks.
4. Include quality control and functional microservices management services in the architecture.
5. Integrate missing functional microservices that provide autonomous execution of business processes, including robotic elements, intelligent management modules, and the Digital Twin of the enterprise.

## 3. Graphical Modeling of the Digital Enterprise Architecture

### 3.1. Problems of Using of Classical Notations

Based on proposals to bring the enterprise architecture to a form suitable for digital transformation, first of all, the question arises about the need to develop project documentation that provides an actual reference plan and a quality control tool for work. The main tool for this purpose is traditionally graphical modeling of architecture. However, taking into account the features described in Section 2, we can conclude that there is some discrepancy between the existing notations of graphical modeling.

Thus, in part of our works [53,54], we were engaged in the development and research of a digital platform for intelligent management of transport and technological processes of open-pit mining operations. At the early stages of designing the architecture of the platform, following ACDM/ATAM methodologies for tradeoff-based architecting of complex large-scale systems [21–23], as well as taking into account the requirements of ISO 25010, we identified the key qualitative attributes of the system—modifiability, scalability, security, and performance, assuming the non-stationarity of the architecture throughout its lifecycle and the need to support its high efficiency. However, due to the complexity of the original object, the use of standard languages for formal graphical description of systems caused great difficulty. Working at the "upper" (conceptual) level of architecting, we were faced with the fact that, on the one hand, it was necessary to use a service-oriented style, i.e., to characterize the horizontal relationship between fundamentally different agents and services (both existing and to be developed), and on the other hand, to show the dynamics of the relationship of such agents and services. Keeping in mind some general considerations on how the architecture could be organized and armed with common sense, we identified the need to first start from the functional purpose of the platform and the entire system as a whole. In other words, we tried to determine the prototype of the future architecture of the entire enterprise system based on the generalization of business processes and operated systems, as well as the inclusion of such a concept as a Digital Twin, limiting it to the main business process—the management of autonomous technical agents in the extraction of mineral raw materials.

Using an agile approach to software development and forming an abstract prototype of the digital enterprise architecture, we identified the main structural and functional elements and selected critical services that need to be checked for viability [21–23,28–30]. Such services primarily included elements of the infrastructure (the platform itself) and functional modules of the Digital Twin. In accordance with this, we developed the MVP (minimal viable product) version of a digital enterprise [53,54], which allowed us to proceed to further research on ways to architect such systems in order to form the most generalized version of the metamodel of the architecture of a digital mining enterprise.

However, the issues of interpreting the resulting model of system architecture at the specification stage in the form of diagrams describing it adequately and not excessively became no less acute than at the initial design stage. First of all, one of the main problems was the lack of a direct connection between graphical representations of business processes,

for which such notations as BPMN, IDEF, etc., can be used, and diagrams of the technical (software) implementation of the system architecture in the form of diagrams in UML notation. The second problem was the redundancy of the resulting diagrams, which are difficult to read and control, which, in our opinion, can lead to errors at different stages of the product lifecycle. Another problem that was mentioned earlier is the lack of convenient diagrams illustrating the dynamics of the structural and functional architecture of the system.

Figure 3 shows a simplified example of using the UML notation for graphical modeling of the dynamic relationship between the structure of services (automated software systems) and microservices in the framework of the task of centralized management of autonomous technical agents (robotic dump trucks and excavators) of an open-pit mining enterprise [4,6]. This scheme, of course, does not reflect all the microservices involved in the process of managing autonomous technical agents, but even in this form, it perfectly demonstrates the disadvantages of traditional notations, namely, the following:

- Redundancy of the number of structural and functional elements and their descriptive component for an unambiguous understanding of the functionality of microservices;
- Redundancy in the number of links between elements for unambiguous understanding of data transfer processes between microservices in order to ensure their operation (and the inability to display links without intersections at all, as required by all notations);
- There is no clear dynamics of relationships between elements that characterize the complex operation of microservices when solving a common business problem.

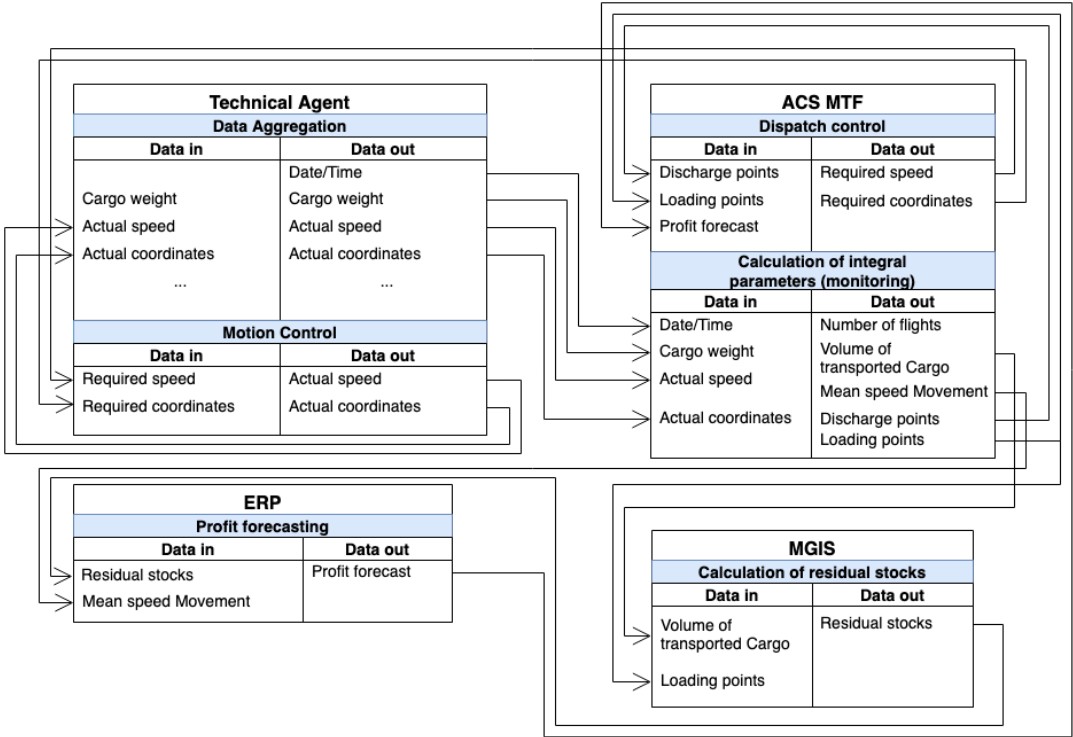

**Figure 3.** Diagram of the structural and functional architecture of solving the problem of centralized management of mining equipment: ACS MTF—automated control system of mining transport facility, MGIS—mining and geological information system, ERP—enterprise resource planning system.

Of course, when creating this diagram, we tried to combine the Component Diagram and the UML ClassDiagram, so that it could be divided into two separate diagrams. However, to meet all these shortcomings, two diagrams would not be enough, and we would have to additionally use an Activity Diagram to illustrate the order of operations of services within the scope of the task, as well as an ER or DFD diagram to illustrate the structure of data and the sequence of their modification to ensure the operation of services.

Thus, no simple and non-redundant way was found for unambiguous graphical modeling of the system architecture using existing notations.

In search of a solution to this problem in the scientific literature, we encountered the Industry 4.0 business process modeling language I4PML (Industry 4.0 Process Modeling Language) [50]. I4PML is essentially an add-on to the language of the formal description of systems—UML (Unified Modeling Language)—in the form of inclusion and allocation of additional designations of Industry 4.0 elements in standard UML structural forms. Due to the fact that the UML itself is already sufficiently abstract, this is what makes it possible to formalize some key elements of Industry 4.0 through it. We recognize the usefulness of unifying such concepts as "Internet of things device", "computing cloud", "act of perception", and "act of execution" at the stage of pre-project and project conceptual development. However, we believe that their application is not universal for all tasks and technologies of Industry 4.0, and in the case of modeling an architectural solution, it is not exhaustive due to the weak formalization of specific technical solutions for organizing information interactions. Another argument against I4PML is the absence of such concepts as "publisher–subscriber relationships" and "data modification scheme". Nevertheless, the language itself is still young, and we see that in the future after accumulating sufficient experience in developing and integrating Industry 4.0 technologies, it will be followed by significant changes (or the appearance of another specialized language).

Thus, we formulated a number of criteria that should be taken into account when compiling a graphical model of a data-centric microservice architecture, and which we used later to solve the problem of this study:

1. Reflect the most complete component composition of the architecture in the form of functional blocks (microservices), while reflecting the sequence of their interactions in solving a business problem.
2. Minimize the number (or completely eliminate) the intersections of communication lines between components in the diagram.
3. Minimize the number of diagrams that provide a complete description of the end-to-end relationship between the components of a business task (functions/microservices), the list and dynamics of data for solving such tasks, as well as a set of data processing methods for solving problems (methods).

### 3.2. DEAL 1.0 Notation

To solve the problem of graphical modeling of the architecture of a digital enterprise, we determined the possibility of using separate UML notation diagrams to describe the technical implementation of microservices, taking into account their slight modification, as well as their hierarchical arrangement into a common family of graphical models as in "4 + 1" or "C4", namely, the following:

- "Functional Diagram" of the enterprise for forming the most general zero conceptual level of architecture. This scheme is not included in the UML, but it is understandable to direct users (industry experts of the enterprise), can be compiled by them, and opens up opportunities for further detailing of individual business task microservices by specialists in software design and development.
- "Process Diagram" describing the structural and functional relationship of services and/or microservices involved in the execution of a single business process. This diagram is based on a UML Class Diagram, a UML Activity Diagram, as well as some ideas from the design of microprocessor electronics circuits. Depending on the complexity of the described business task, this diagram can be depicted immediately for microservices, or in the form of two consecutive hierarchically linked diagrams—for services and for microservices.
- "Microservice Architecture Diagram" illustrating the pre-program view of a microservice, highlighting all the basic methods necessary to implement its functional task, as well as its actual physical deployment location. This diagram is formed on the basis of a Component Diagram and a UML Class Diagram.

- "Pipeline Diagram" illustrating the relationship between two microservices on the principle of data publisher and subscriber, as well as the direct data structures that supply and receive microservices within their functionality. This diagram is also based on a UML Class Diagram and some elements from ER notation diagrams.

We named the proposed approach to design a digital enterprise architecture model "DEAL 1.0"—Digital Enterprise Architecture Language [55] (Figure 4).

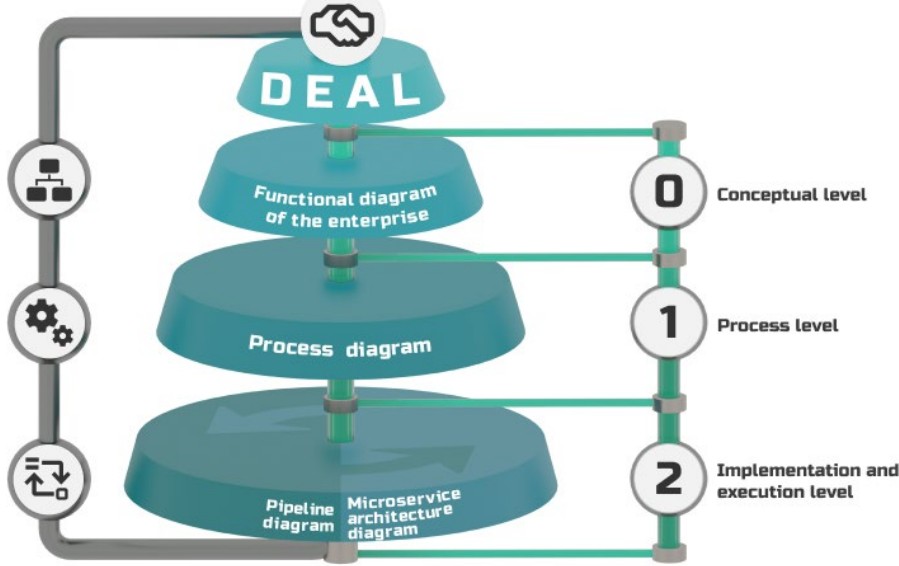

**Figure 4.** Composition and relationship of digital enterprise architecture modeling diagrams "DEAL 1.0" (Digital Enterprise Architecture Language).

The key basis of the proposed notation of graphical architecture modeling is the formalization of knowledge about processes, operations (elementary indivisible functional tasks—microservices), methods (used to solve problems), the sequence of interactions of components, information about structures, and dynamics of data changes (Figure 5).

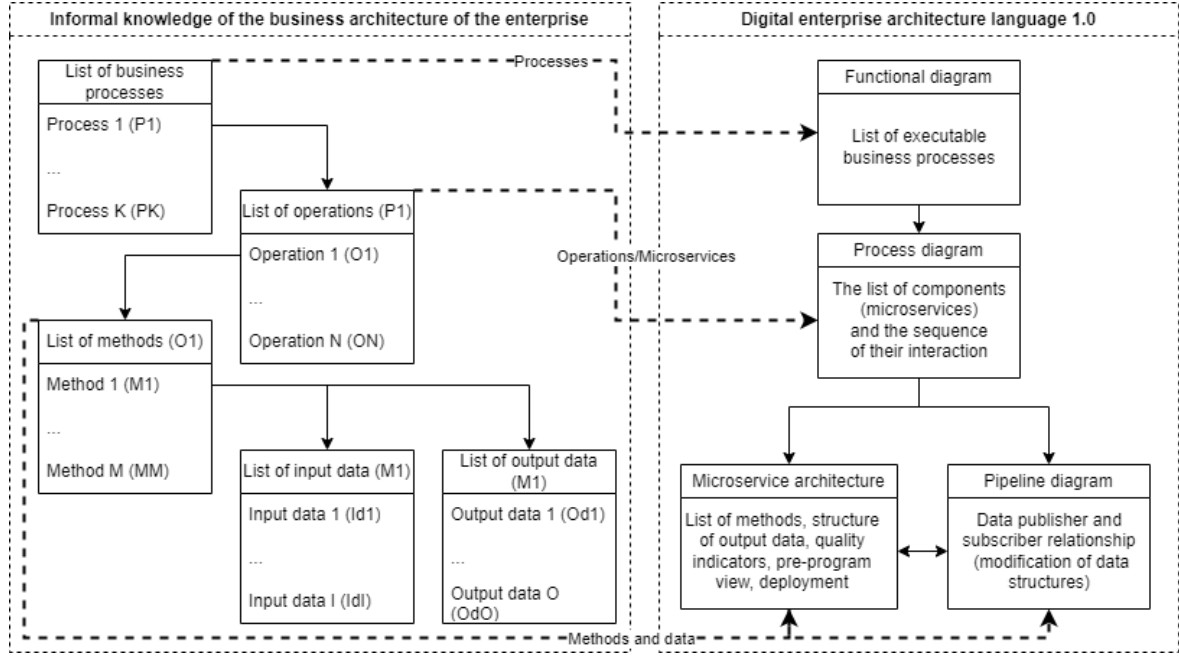

**Figure 5.** Formalization of the relationship between information about the business architecture of the enterprise and the DEAL 1.0 notation.

### 3.3. Conceptual Level (0): Organizational and Functional Diagram

The zero "conceptual" level of our approach to modeling the architecture of a digital enterprise involves using a fairly simple and widely used organizational and functional diagram. We do not think it makes sense to explain how it is built, but we would like to highlight some key aspects related to the reasons for its use:

1.　Modern enterprises are complex systems with a difficult to formalize connection between business tasks. The description of the full list of such tasks, the sequence of their implementation, quality criteria, data, and methods of solution can be described only with the participation of specialists of the enterprise itself. However, employees of enterprises do not always have knowledge of graphical modeling notations, so the initial design point with the minimum entry threshold must be found—i.e., the simplest diagram possible.
2.　Moreover, as a rule, each enterprise already has an organizational model that identifies key decision makers and departments involved in business processes. Turning an organizational model into a functional one is quite simple and does not require specific knowledge and skills in developing system modeling diagrams—it is enough to replace the designations of decision makers and departments in the organizational model with business tasks that are central to their activities.

Figure 6 shows an example of such an organizational and functional diagram applied to a mining enterprise. In this diagram, the intent is to exclude branches related to the management of financial, economic, and human resources activities in order to illustrate exactly the key technological processes and highlight the business task that was shown in Figure 3.

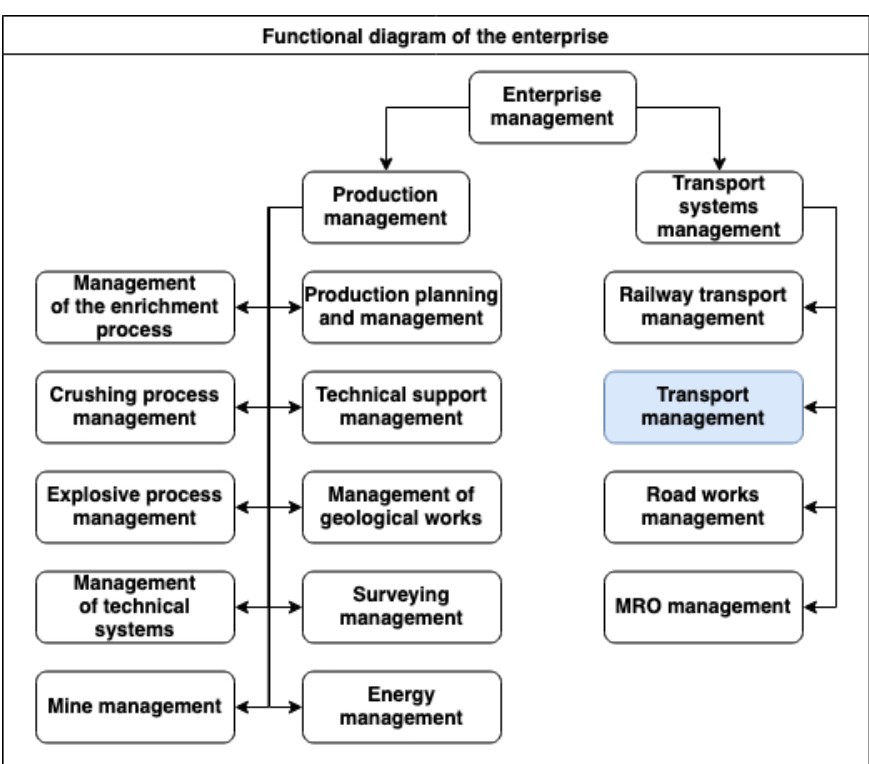

**Figure 6.** Conceptual Level (Level 0)—organizational and functional diagram of the enterprise.

### 3.4. Process Layer (1): Process Diagram

Following our notation, it is then necessary to "fall through" into each of the tasks to create diagrams describing which elementary (indivisible) operations they consist of (microservices) and what is the order of these operations (interaction of microservices). Figure 7 shows such a diagram for the task of centralized management of robotic agents.

For ease of perception, individual microservices, which in the architecture of a digital enterprise imply an independent existence with a weak connection with other microservices, were grouped by control contours—which agents and systems (services) they belong to in the existing enterprise architecture. Keep in mind that such a grouping and simultaneous display of services and microservices is more undesirable than optional. If it is impossible (difficult) to move from a Functional Diagram to a Process Diagram indicating microservices, it is worthwhile to form in a hierarchical relationship first a diagram of the sequence of connecting services (i.e., group microservices by agents and control contours), and then proceed to its detailed form, decomposing each service into separate independent indivisible functional microservices.

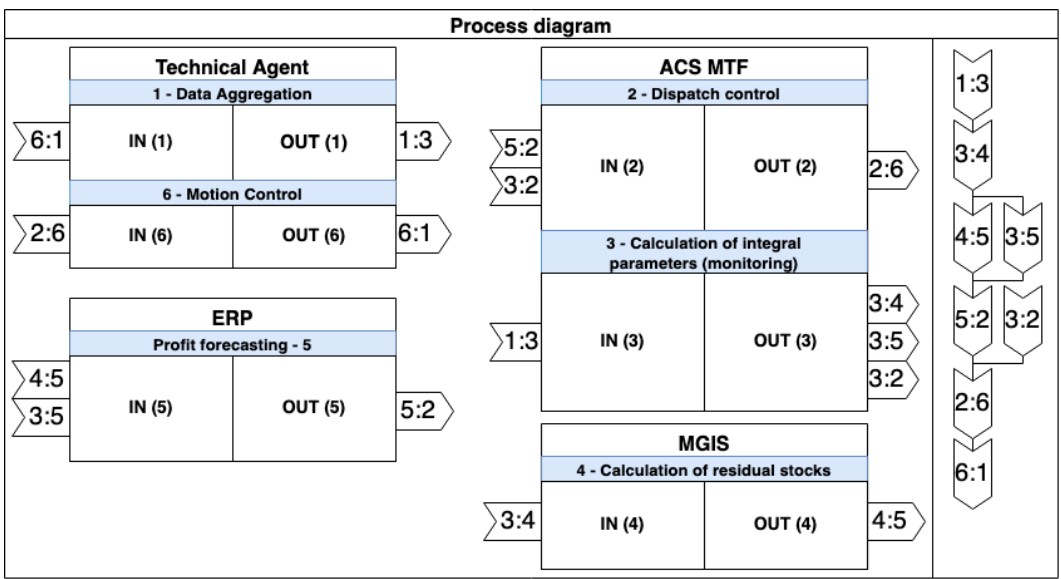

**Figure 7.** Process Level (Level 1)—Process Diagram.

In this type of diagram, the designer assigns a unique identification number to all microservices. For example, the microservice "Data Aggregation" of the agent "Technical Agent" (robotic dump truck) is assigned the number 1. At the same time, we do not see the need for a strict order of assigning identification numbers, since they serve exclusively to indicate the connection between microservices, and the sequence of functions performed during their information interaction. Keep in mind that different business tasks can (and should) use the same microservices, because the service-oriented architecture implies the inheritance of functions, as a result of which the identification number for each microservice must be unique and use the same for diagrams of different tasks.

Each microservice has an external input (IN) and output (OUT) for receiving and transmitting data, which directly correlate with the concepts of "publisher" and "subscriber". To avoid redundancy of connections and their intersection, we propose to indicate the connection of two services in the form of a flag: data from the external output of one microservice should go to the input of some other microservice, which converts it into new data and sends it to its external output for transmission to the input of the next microservice (s), etc. At the same time, the flag itself always indicates the identification number of the microservice from which the information comes, and on the right number of the microservice to which the information is sent. We borrowed and reinterpreted this approach to marking connections from the field of designing microprocessor boards, where the problem of readability of element connections has been known for quite a long time. In particular, when designing circuits for microprocessor boards, the "end-to-end" connection of elements is used to avoid intersections of conducting tracks, which in some ways is similar to the idea of a weakly connected end-to-end connection of microservices—"pipelines".

In view of the possible disordered assignment of identification numbers to microservices and the determination of their sequential interaction, we propose to depict the sequence of microservices operations within the framework of a simulated business problem in the right part of this diagram. This process part is intended to show the order in which the functions of microservices are performed and what information should be transferred from one microservice to another at what point in time. At the same time, the order of operations of microservices is focused exclusively on data, i.e., the data-centric architecture is traced, since in the left part of the diagram, a specific microservice of one type can be changed to another microservice (another software vendor) that provides better performance indicators (considering the performance of identical functionality).

Another feature of this diagram is the fact that microservices at this level are an abstract designation of the functional task being performed and can be implemented both in the form of specific software and in the form of an operation performed by a specialist—an employee of the enterprise in manual mode. One unaccounted-for-problem at the moment is the display of a list of already-existing microservices. For this purpose, it is possible to use a frame with table entries, numbering, names, and functions of microservices, generated automatically.

Thus, the "alphabet" of the Process Diagram is as follows:

1. The diagram is divided into two parts: the left structural part shows the services/microservices involved in the implementation of the business process, as well as their relationships with other services/microservices to perform their own functions; the right part shows the order in which services/microservices perform functions during the implementation of the business process.
2. "Service"/"Microservice"—indicated as a rectangle divided into three parts: the upper part contains the name of the service/microservice, the lower left part serves as an " external input "(IN) for subscribing to data and contains the identification number of the service/microservice, and the lower right part serves as an "external output" (OUT) for publishing (publishing) data.
3. "Flag"—indicates the connection in the left part of the diagram between two services/microservices, containing in the left part the number of the service/microservice from which the information comes, and on the right the number of the service/microservice to which the information comes.
4. "Communication line"—indicates the connection on the right side of the diagram between the "checkboxes" to determine the sequence or parallelism of microservices. The absence of an input communication line in time implies asynchronous, i.e., independent of the previous operation and mode of operation of the microservice.

*3.5. Implementation and Execution Level (2): Microservice Architecture Diagram and Pipeline Diagram*

The next step in modeling the architecture of a digital enterprise is to create diagrams of microservice architectures and Pipeline Diagrams. The order of development of these diagrams does not matter; however, they assume both a direct hierarchical relationship with the Process Diagram and a direct horizontal relationship with each other. To go to the diagram of the microservice architecture, it is necessary to "fall" into each separate microservice, and to go to the Pipeline Diagram, it is necessary to "fall" into each separate connection between two microservices.

The Microservice Architecture Diagram (Figure 8) is a combination of a Component Diagram and a UML Class Diagram. In the upper "component" part, the name of the physical location (deployment) of the microservice (in our case, it is a "Technical Agent") is displayed, as well as the necessary technical information about the methods, tools, and conditions for performing the placement of the microservice—this can be, for example, the technical characteristics of computing devices.

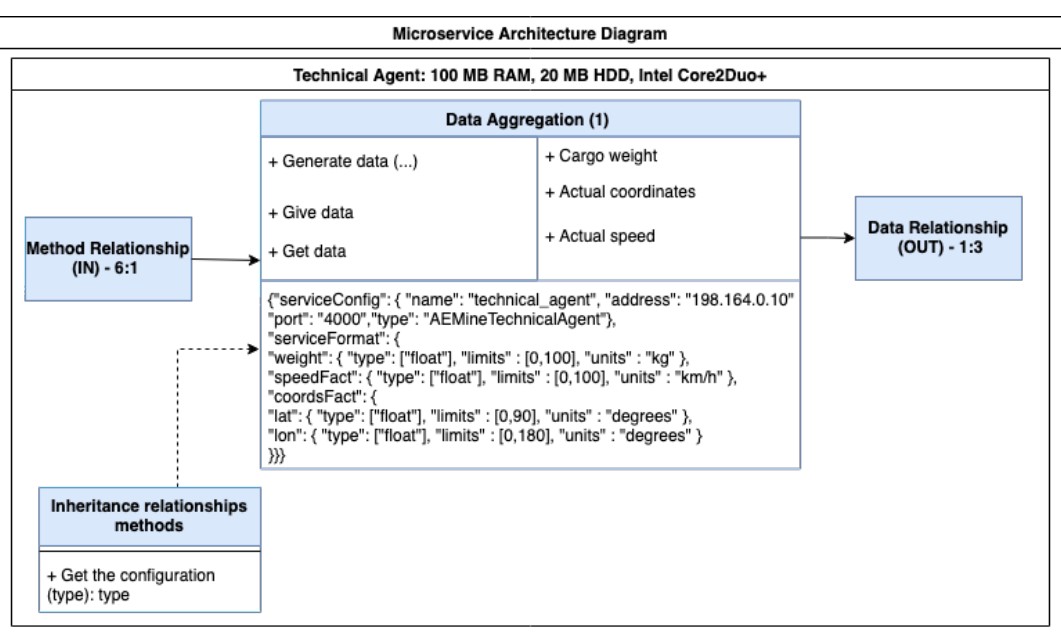

**Figure 8.** Implementation and execution layer (2): Microservice Architecture Diagram.

The main part of this chart displays the following:

1.  "Microservice"—a vertical rectangle divided into four parts: in the upper part, the microservice ID and name are displayed; in the middle left part, the microservice "methods" are displayed; in the middle right part are "data" for external output; in the lower part, "configuration" is displayed:

    -   "Methods"—a set of simple (computational) operations that a given microservice operates to implement a given function and that can be implemented either using program code or manually by enterprise specialists;
    -   "Data"—information that the microservice produces based on the results of performing its own functions (methods) and that is sent to an external output;
    -   "Configuration"—a software representation of the microservice structure that reflects a set of service parameters of the microservice (ID, address, port, type, state, quality of work, etc.)  for the broker to organize its interaction on the integration bus with other microservices, as well as meta-information of the structure of its external input (IN) and external output (OUT).

2.  "Links"—horizontal rectangles with a connecting line that indicate the following:

    -   "Method links" (IN)—links associated with other microservices that transmit information to microservice methods for implementing its functions;
    -   "Data links" (OUT)—links associated with an external output for publishing and transmitting information to other microservices;
    -   "Inheritance links"—links associated with shared inherited methods from the enterprise service library that are necessary for the operation of the microservice (for example, a request for configuration, setting up data-sending schemes, determining system time, etc.).

The DEAL Pipeline Diagram is a combination of the UML Class Diagram and the ER Diagram principles. The Pipeline Diagram (Figure 9) shows a direct connection between the output (OUT) and input (IN) of two microservices—"pipeline". The microservice from which information is received (OUT) in this connection should be shown on the left, and the microservice to which information is received (IN) should be shown on the right. This diagram indicates the following:

1. "Microservice"—a horizontal rectangle divided into three parts: the upper part contains the identification number and name of the microservice; the middle part contains the "list of data"; the lower part contains the program view of the "data package".
2. "Data list"—a block containing the name of data sent or received by microservices during the implementation of their functions, as well as human-readable data characteristics: number type, measurement range, unit of measurement, etc.
3. "Data package"—formats of program representation of data at the output and input of microservices, necessary for subsequent data transformation by the broker on the integration bus in order to actually implement information transfer between microservices in accordance with their capabilities and needs.

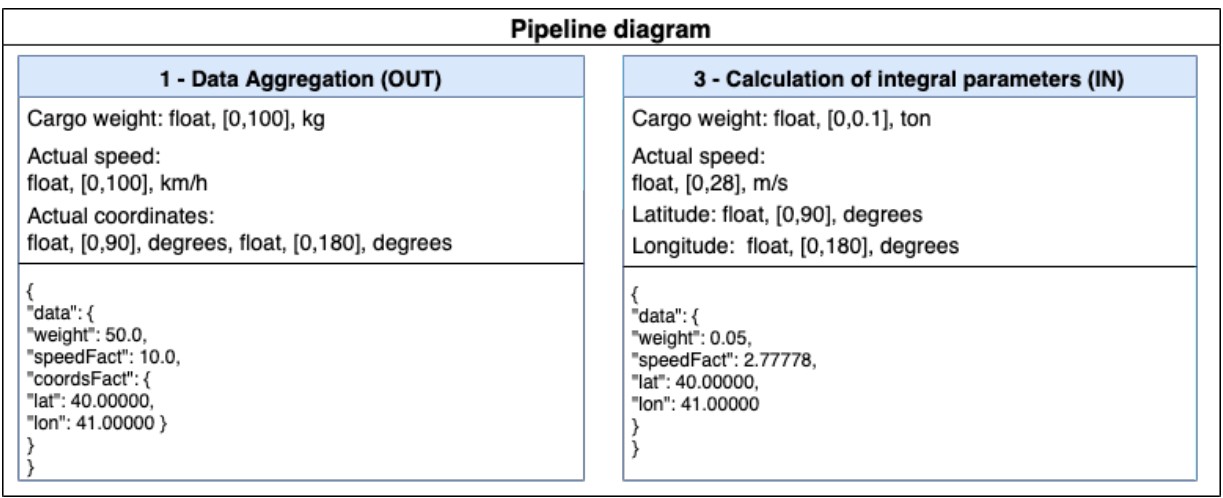

**Figure 9.** Implementation and execution level (2): Pipeline Diagram.

The Microservice Architecture Diagram and the Pipeline Diagram suggest the possibility of switching from one to the other; they can be understood by both industry specialists of enterprises and software developers; in fact, they allow one to move to the direct software implementation of the digital enterprise architecture and, in our opinion, fully meet the ideology of data-centric microservice system architecture.

## 4. An Example of the Implementation of a Digital Enterprise Architecture

*4.1. DEA 1.0—Digital Enterprise Architecture Metamodel*

In accordance with the above proposals for implementing the digital transformation of the enterprise, as well as based on the generated diagrams in the DEAL 1.0 notation, we developed a functional software metamodel of the architecture [56], which is a prototype version of an open-source digital mining enterprise and has the following characteristic features (Figure 10):

1. Due to the complexity of predestination and the actual impossibility of processing operated systems that are directly involved in the main technological process—mineral extraction in the developed model—key agents and systems are presented not in microservice form, but in the form of services—enlarged modules. These modules are as follows:

    - AETechnicalAgent—technical agents, which mean robotic machines of the mining transport complex (dump trucks and excavators) operating inside the technological environment, collecting and providing primary data;
    - AEAHS—dispatching system for the mining transport complex (AHS ACS), which is responsible for monitoring the condition and managing technical agents;

- AEMGIS—mining and geological information system (GGIS), which is conventionally responsible for determining the geostructure of a quarry and estimating the reserves of a field;
- AEERP is an enterprise resource planning and management system that performs high-level production management tasks.

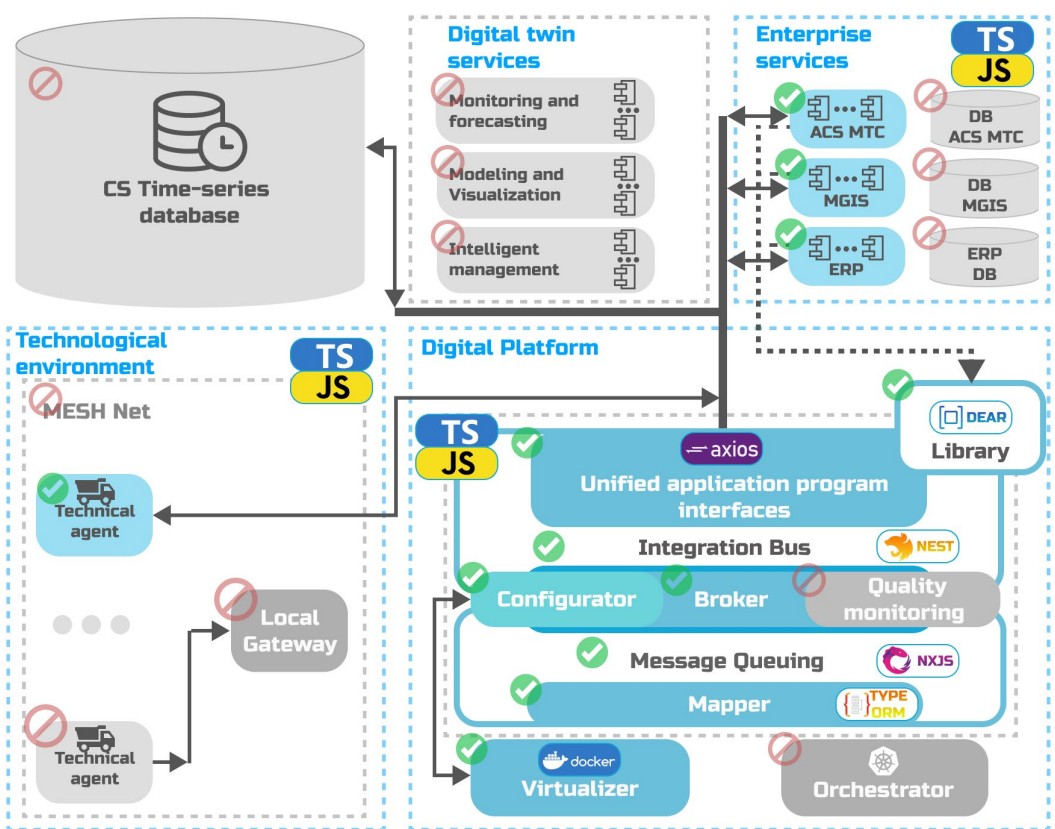

**Figure 10.** The current meta-model of the digital enterprise architecture DEA 1.0.

All the listed agents and systems in the developed metamodel are functioning components and directly carry out conditional (incomplete) information exchange during the implementation of the business process. At the same time, each of the agents and systems has knowledge about their own functionality (configuration), which they report to the corresponding service components of the model in order to publish or receive data that are necessary or obtained during the implementation of their own functions. The proposed model assumes the following unified types of messages from services:

- AvailableData—announcement about the possibility of publishing data;
- RequiredData—declaration of the list and structure of required data;
- GetStatus—warning about the possibility of receiving data;
- SetConfig—request to install the configuration and service;
- GetInConfig—request to obtain the configuration of the input data structure;
- GetOutConfig—request to obtain the configuration of the outgoing data structure;
- Command—not a deterministic command.

In accordance with the type of agent or system, each of the services publishes data that are produced based on the results of its work and aggregated in the platform part of the model according to the corresponding data topics (topics), to which other services can connect to receive such data:

- TechnicalAgentData—the topic declared by the technical agent;
- AHSData—a topic declared by the dispatching system for the mining transport complex;

- MGISData—a topic declared by the mining and geological information system;
- ERPData—topic declared by the enterprise resource planning and management system.

Each of the services (agents and systems) also has information about its own qualitative state of functioning, the program view of which is currently implemented as follows:

export enum ServiceStateTypes {

- Normal—normal mode of operation;
- NotSet—not configured;
- CorruptedModule—there are obstacles to normal operation;
- Fatal—critical error}.

2. The platform part of the AEBus architecture (Digital Platform) AEBus is a set of service components and libraries that directly ensure the effective interaction of functional services (agents and systems) and includes the following:

  - The "Integration Bus" service, which is actually implemented unified software interfaces for applications (agents and systems), as well as a single library of methods that agents and systems access to connect to a common message transmission scheme.

The library of application programming interfaces contains several types of possible connections—for services (existing and non-recycled systems) and for microservices (for new or recycled systems).

The ServiceApiInterface implements a typical CRUD (create, read, update, and delete) concept, which allows one to create, read, update, or delete data in the system. This interface extends the AppService class (or another similar technology for performing asynchronous/high-load operations). The interface can be used for operated systems without making significant changes to their software implementation.

The MicroServiceApiInterface unifies interactions with any microservice by providing a single point of access to data, configurations, and entity types. This interface can (and should) be used for all existing services or those being developed and included in the architecture of microservices.

The ServiceState interface provides internal interactions of the service components of the platform for managing functional services and microservices.

- The "Broker, message queue, and mapper" service consists of several separate microservices that perform the functions of organizing communication between services (creating a pipeline), listening for data requests and messages about publishing data, forming message queues, manipulating (adding, deleting, etc.) messages in queues, and modifying the data contained in messages to bring them up to date. They are derived from the incoming data structure formats from some microservices to the required formats of others.

The architecture metamodel including methods and tools of the library is available on the GitHub repository in open access under the MIT license, available for download and free use: https://github.com/kinozal1/DEAMetamodel/ (accessed on 29 November 2022).

*4.2. Description of Functional Modeling of DEA 1.0 Operation*

The process of functioning of the developed software metamodel of architecture, which is named Digital Enterprise Architecture 1.0 (Digital Enterprise Architecture), is carried out as follows:

1. The services of the digital platform—AEBus—are being initialized.
2. AEBus services are ready to work and are waiting for microservices to be connected.
3. Initialization of microservices AETechnicalAgent, AAHS, AEERP, and AEMGIS is taking place. The microservice knows its initial configuration and the address of the platform broker. After initialization, the microservice is registered, and it reports its configuration according to the following structure:
   *export class RegisterService {*

*name: string; // service name*
*address: string; // address*
*port: number; //*
*port type: ServiceTypeTypes; // type }*

4. After registration, the platform broker stores a view of each of the microservices in the following format:
   *export class ServiceInfo {*
   *id: number; // id number*
   *name: string; // name*
   *available: boolean; // availability*
   *instance: any; // instance*
   *config: any; // configuration*
   *quality: number; // quality*
   *type: ServiceTypeTypes; // type*
   *port: number; // port*
   *address: string; // address }*
   Further, the entire process is carried out in asynchronous mode in a cyclic form.

5. The AETechnicalAgent service registers an event about publishing data:
   *{ topic: TopicTypes.TechnicalAgentData, // topic name*
   *event: MessageTypes.AvailableData, // with communication about the publication of data (list and its structure)*
   *sender: ServiceTypeTypes.AETechnicalAgent, // name of the message sender }*

6. The broker adds an event to the queue and checks for subscribers to the current TechnicalAgentData tag.

7. The AEAHS service registers data necessity events (a) and, after receiving and processing them, data publication events (b) and (c):

   (a)  *{ topic: TopicTypes.TechnicalAgentData, // topic name*
        *event: MessageTypes.RequiredData, // request to receive data (list and its structure)*
        *sender: ServiceTypeTypes.AEAHS, // name of the sender of the message }*

   (b)  *{ topic: TopicTypes.MGISData, // name of the topic*
        *event: MessageTypes.AvailableData, // message about publishing data (list and its structure)*
        *sender: ServiceTypeTypes.AEAHS, // name of the sender of the message }*

   (c)  *{ topic: TopicTypes.ERPData, // topic name*
        *event: MessageTypes.AvailableData, // message about publishing data (list and its structure)*
        *sender: ServiceTypeTypes.AEAHS, // name of the sender of the message }*

8. The broker adds events to the queue and checks for matches in the publisher and subscriber queues. It finds a match for available data from the AETechnicalAgent microservice (technical agent) and the need for data acquisition by the AEAHS service (SCC dispatching system). It initiates the receiving of data from the AETechnicalAgentthe AETechnicalAgent Data queue, converting the data to the required format, and transmitting them to the AEAHS service.

9. The AEMGIS service registers data necessity events:
   *{ topic: TopicTypes.MGISData, // name of the topic*
   *event: MessageTypes.RequiredData, // message for receiving data (list and its structure)*
   *sender: ServiceTypeTypes.AEMGIS, // name of the sender of the message }*

10. The broker adds an event to the queue and checks for matches in the publisher and subscriber queues. It finds a match with the AEAHS service. It initiates the receiving of data from AEAHS, converting and transmitting data to the AEMGIS service.

11. The AEERP service registers data necessity events:
    *{ topic: TopicTypes.ERPData, // topic name*
    *event: MessageTypes.RequiredData, // message for receiving data (list and its structure)*
    *sender: ServiceTypeTypes.AEERP, // name of the sender of the message }*

12. The broker adds an event to the queue and checks for matches in the publisher and subscriber queues. It finds a match with the AEAHS service. It initiates the receiving of data from AEAHS, converting and transmitting data to the AEERP service.

Figure 11 shows a visualization of the operations of the metamodel in accordance with the process described above.

```
27.10.2021 04:54:17.000 | broker running on port 4000

27.10.2021 04:54:23.248 | AETechnicalAgentData has been registered on Bus!

27.10.2021 04:54:23.251 | Event from AETechnicalAgentData, with TechnicalAgentData topic

27.10.2021 04:54:27.554 | AEAHS has been registered on Bus!

27.10.2021 04:54:27.555 | Event from AEAHS, with TechnicalAgentData topic

27.10.2021 04:54:27.564 | Event from AEAHS, with TechnicalAgentData topic

27.10.2021 04:54:27.566 | Event from AEAHS, with MGISData topic

27.10.2021 04:54:27.567 | Event from AEAHS, with ERPData topic

27.10.2021 04:54:27.575 | Data mapped and wrote to AEAHS - {"sender":{"id":"34067","name":"Mine Truck 86","adress":"55.190.208.246"},"timestamp":"16337753205
t":"17","lat":"12.4891","lon":"-160.2310"},{"weight":"122","speedFact":"22","lat":"9.1574","lon":"63.9092"},{"weight":"77","speedFact":"19","lat":"-44.1548",
2"},{"weight":"94","speedFact":"14","lat":"-89.3840","lon":"-72.2020"},{"weight":"72","speedFact":"10","lat":"-44.5920","lon":"112.3223"},{"weight":"37","spe
","lat":"23.5357","lon":"-118.9664"},{"weight":"24","speedFact":"38","lat":"70.1994","lon":"-71.5791"},{"weight":"106","speedFact":"31","lat":"21.6421","lon"

27.10.2021 04:54:31.367 | AEERP has been registered on Bus!

27.10.2021 04:54:31.368 | Event from AEERP, with ERPData topic

27.10.2021 04:54:31.368 | Event from AEERP, with ERPData topic

27.10.2021 04:54:31.370 | Data mapped and wrote to AEERP - {"sender":{"id":"34067","name":"Mine Truck 86","adress":"55.190.208.246"},"timestamp":"16337753205

27.10.2021 04:54:34.264 | AEMGIS has been registered on Bus!

27.10.2021 04:54:34.265 | Event from AEMGIS, with MGISData topic

27.10.2021 04:54:34.265 | Event from AEMGIS, with MGISData topic

27.10.2021 04:54:34.267 | Data mapped and wrote to AEMGIS - {"sender":{"id":"34067","name":"Mine Truck 86","adress":"55.190.208.246"},"timestamp":"1633775320
"},{"id":100,"tech_agent_id":"34067","lat":"9.1574","lon":"63.9092"},{"id":100,"tech_agent_id":"34067","lat":"-44.1548","lon":"127.0429"},{"id":100,"tech_age
7","lat":"-89.3840","lon":"-72.2020"},{"id":100,"tech_agent_id":"34067","lat":"-44.5920","lon":"112.3223"},{"id":100,"tech_agent_id":"34067","lat":"34.5056",
8.9664"},{"id":100,"tech_agent_id":"34067","lat":"70.1994","lon":"-71.5791"},{"id":100,"tech_agent_id":"34067","lat":"21.6421","lon":"-168.0901"}]}
```

**Figure 11.** Fragment of visualization of the metamodel operation.

The main idea of the proposed approach using a unified library is the possibility of flexible integration of microservices and organization of their information interaction in the current architecture according to common standards (interfaces) without significant changes in the software implementation of the systems used in enterprises. The implementation of libraries also implies storing unified templates of microservice abstractions, which provide the possibility of further expanding the existing architecture to include missing microservices that ensure enterprise autonomy. Each object of microservice interaction (interfaces, data objects, data modification logic) is stored in an external generalized library and, if necessary, is connected by each microservice separately.

## 5. Discussion

Based on the results of the work carried out, the following was achieved:

1. The analysis of the generalized structural and functional architecture of business processes of modern industrial enterprises was carried out, within which the key features that need to be taken into account or changed for the implementation of their digital transformation were identified.
2. Specific iterative steps necessary for the implementation of digital transformation of enterprises in terms of bringing the architecture of business processes to a data-centric microservice form were proposed.
3. The rationale for the need to develop a graphical modeling language for designing systems with a data-centric microservice architecture that meets the requirements of Industry 4.0 was given.

4.  A method for formalizing such a language—Digital Enterprise Architecture Language 1.0—was proposed.
5.  A formal representation of the unified reference metamodel of the digital enterprise architecture DEA 1.0, which can provide autonomous execution of business processes, was given.
6.  An example of a software implementation of such a metamodel was shown, as well as an example of experimental modeling of its operation, in which the functional viability of the chosen approach was determined.

The proposed metamodel of the digital enterprise architecture DEA 1.0, in general, corresponds to the data-centric microservice approach, since it contains all the necessary service components responsible for organizing the information interaction of microservices and takes into account the development features of missing promising microservices, including those related to the Digital Twin of the enterprise. At the same time, at this stage, the model does not implement a Time-Series Database (TDBM), which is necessary to improve the performance of large data transfer processes. However, the proposed approach considers this feature and, in the future, the TDBM can be easily implanted into the existing architecture. In the future, it is planned to expand the developed libraries and refine the service components responsible for monitoring the quality of microservices, orchestration, and virtualization, as well as to study the functioning of the platform part of the model under a high-performance load in order to refine the architecture.

Regarding the DEAL 1.0 notation, it is quite difficult at the moment to talk about its full compliance with the challenges of designing Industry 4.0 systems. However, when developing it, we focused on the following key points:

1.  The possibility of application for various subject areas (industrial enterprises). Despite the fact that this paper provides an example exclusively for mining, the alphabet of the language retains the traditional principles of graphical modeling and, in our opinion, is sufficiently unified, so that most of its elements can be applied to the design of a system from another subject area (for example, the construction sector).
2.  The need to find a compromise between industry specialists and system developers. Thus, the organizational and functional diagram is easy to understand for both parties and is the starting point for design. Further diagrams, of course, have their own specifics, but with the formalization of the sequence of operations (Process Diagram) or, for example, the representation of a set of operations and data obtained from the results of their work (Microservice Architecture Diagram), they can be specified as in the contract, from the point of view of software development, form, i.e., performed in manual mode, and in directly pre-programmed form.
3.  With significant refinement of the notation, in particular the creation of functions for the automatic generation of a list of microservices, it will be possible to ensure control of the architecture design in terms of both the correctness of the formation of microservices themselves (strict decomposition to the simplest operations) and compliance with the rules on the inadmissibility of duplication of functions in the architecture of the entire system.
4.  Reduction of redundancy of elements and links, visual representation of links. The use of "flags" in the most voluminous scheme—the Process Diagram—in our opinion significantly simplifies the perception of both the sequence of connections and connections between microservices in general. The reduction of the elements, in this case, can be achieved by controlling the inadmissibility of duplication of microservices as required by the microservice paradigm. In other words, when creating a diagram for the next process, we do not create new entities (microservices) but take them, if available (corresponding to the functional purpose), from a common "library".
5.  The possibility of a visual joint representation of the elements of a data-centric microservice architecture that displays both the dynamics of the structural and functional relationship of components and data modification schemes used in the course of information exchange between such components. We do not claim that DEAL 1.0 is

fully ready to become a key notation for Industry 4.0, but we offer one of the possible options for a new design method that formalizes key approaches to the phased construction of autonomously functioning digital enterprises.

It is also worth noting that the proposed method of graphical modeling does not reflect, in our opinion, another important (third) level of technical functioning of the system, which describes the organization of service components and the order of their interaction with functional microservices. In the future, we plan to continue work in this direction, including the software implementation of the alphabet in the form of an embedded library in open source graphical modeling tool environments.

**Author Contributions:** Conceptualization, S.D., U.R. and E.K.; methodology, S.D.; software, U.R. and E.K.; validation, S.D., U.R. and E.K.; formal analysis, I.T.; investigation, S.D.; resources, E.K.; data curation, U.R.; writing—original draft preparation, S.D.; writing—review and editing, I.T. and E.K.; visualization, U.R. and E.K.; supervision, I.T.; project administration, I.T.; funding acquisition, I.T. All authors have read and agreed to the published version of the manuscript.

**Funding:** This research received no external funding.

**Institutional Review Board Statement:** Not applicable.

**Informed Consent Statement:** Not applicable.

**Data Availability Statement:** The DEA 1.0 metamodel, including methods and tools of the library, is available on the GitHub repository in open access under the MIT license, available for download and free use: https://github.com/kinozal1/DEAMetamodel/ (accessed on 29 November 2022).

**Acknowledgments:** This article contains materials from two previously published works, originally published in Russian by the publishing house, "Mining book", in the journal, "Mining Informational and Analytical Bulletin" (Deryabin, S.A., Kondratev, E.I., Azar ogly, R.U., Temkin, I.O. Digital Mine architecture modeling language: Methodological approach to design in Industry 4.0. Mining Informational and Analytical Bulletin, 2022, 2022, pp. 97–110, https://10.25018/0236_1493_2022_2_0_97 [55] and Deryabin, S.A., Azarogly, R.U., Kondratev, E.I., Temkin, I.O. Metamodel of autonomous control architecture for transport process flows in open pit mines. Mining Informational and Analytical Bulletin, 2022, 2022, pp. 117–129, https://10.25018/0236_1493_2022_3_0_117 [56]). The current materials have undergone re-conceptualization, generalization, and correction of key theses and results, as well as a significant expansion of the content. The translation was prepared by the authors of this and two previous publications at their own expense. The authors of this work are the authors of previously published materials and full copyright holders, and no additional permissions are required to publish the article in its current form.

**Conflicts of Interest:** The authors declare no conflict of interest.

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
