# Peer review of "Models and Methods of Designing Data-Centric Microservice Architectures of Digital Enterprises"

_informatics, doi:10.3390/informatics10010004_

Round 1

Reviewer 1 Report (Previous Reviewer 1)

1. The title of the article does not match its content. It does not present an exhaustive analysis and development of new methods and tools for designing the digital architecture of enterprises. A private subject area is considered. The conclusions cannot be automatically extended to larger enterprises.

2. The article seems as a review and does not contain the development of new relevant methods, approaches and algorithms that could be applied in practice for solving problems of optimizing production processes.

3. A specific class of processes that the authors are trying to optimize as part of digital transformation has not been analyzed or singled out. Accordingly, clear optimization criteria have not been developed that could be verified during the review. It cannot be confirmed that the author's concept has a positive effect.

4. An experimental verification of the approach proposed by the authors in solving real practical problems has not been presented. Conclusions has not been confirmed by experiment.

5. There is no significant comparison between the approach proposed by the authors and other well-known methods of Industry 4.0 digital transformation.

6. A particular case of using the modeling language developed by the authors is considered (Section 4.1; Figure 10). There are no explanations of how this method can be generalized to model systems of other classes within the framework of the declared transformation. All examples are particular. No specific task for a real enterprise has been completely solved, or is not described in the article.

7. Duplicate numeration at pos. 622 and 707: 4.1. DEA 1.0 - digital enterprise architecture metamodel (622), 4.1. Description of functional modeling of DEA 1.0 operation (707).

8. GitHub repository is not containing any declared in lines 703-706 and 867-870 detailed descriptions.

9. The choice of a subset of UML and the rejection of other proven and standardized notations such as IDEF are not justified. This is a consequence of the absence in the work of clearly formulated differences between the considered class of systems and other previously known ones. The introduction of new technical elements into the system does not mean a new conceptual level of design.

10. Model -- a system of postulates, data, and inferences presented as a mathematical description of an entity or state of affairs (Merriam-Webster). Authors not presented any mathematical descriptions which could be classified as the model.

11. The choice of microservice architecture as the base one is not justified. Moreover, for universal interaction with real objects, other standards are used, for example, the IEEE family.

Thus, the updated version of the article does not contain a significant scientific component or practical results that could be of interest to researchers working in similar areas, and should be significantly improved in accordance with the comments.

Author Response

Dear reviewer!

Due to the fact that the responses to your comments turned out to be quite large, we attach them as an attached file.

With best wishes,
Authors

Reviewer 2 Report (Previous Reviewer 2)

I am satisfied that the authors have adequately addressed the comments I have mentioned in my initial review of this text.

Author Response

Dear reviewer!

Thank you for your work, careful study of the article and comments on improving its quality.

We found your comments interesting, but we have to disagree with them.

Remark 1:

I am satisfied that the authors have adequately addressed the comments I have mentioned in my initial review of this text.

Response to remark 1:

We express our sincere gratitude for the comments provided earlier to improve the quality of the article!

We are glad that you found our work interesting and worthy of readers' attention.

With best wishes,

Authors!

Reviewer 3 Report (New Reviewer)

This paper presents an approach to designing a digital enterprise architecture. The problem statement is understandable. The authors state that "A relatively new direction can be attributed to the concept of a Digital Twin of an enterprise". However, this Digital Twin is probably not a 100% novel idea, and therefore the contribution does not seem 100% original.  

The paper is rather lengthy; however, it probably lacks a clear relationship between the multiple notations (incl. DEAL, UML, functional, process etc.) the authors use. Additionally, as some of these notations are non-standard ("we tried to combine the Component Diagram and the UML ClassDiagram"  - p.10), this complicates the process of comprehending their overall i.e., high-level, design ideas and plans. As the artifacts in these diagrams often seem vague by themselves and loosely related to each other, we would recommend the authors to develop a clearer research statement both in prose and as a high-level process diagram to make these subtle relations more obvious.

This 24-page long paper lacks a dedicated Conclusion Section; therefore, we would recommend adding an explicit Conclusion prior to Discussion, and verifying that each result obtained clearly corresponds to a certain part of the problem statement.

Overall, this paper requires a thorough revision in terms of research methodology. Therefore, we would like to draw the attention of the authors to a few issues to address while revising the paper; these are detailed below. 

1. Fig.1 is probably designed/developed elsewhere; however, this is not referenced properly. Therefore, we would recommend double-checking the figures and referencing them appropriately if/where applicable.      

2. Verification part needs improvement, as the example(s) presented are not examined in terms of business values and/or critical quality attributes of the system to be designed. Instead, authors suggest an MVP "which allowed us to proceed to further research on ways to architect such systems in order to form the most generalized version of the metamodel of the architecture of a digital mining enterprise". However, it is uncertain how this MVP is related to "the most generalized version of the metamodel". Moreover, such business values and/or mission-critical system quality attributes are not clearly identified and extensively covered.

3. The paper deals with system design; however, it does not present a clear architecture diagram, even in its software part. Therefore, we would recommend adding (i) data view, (ii) process view, and (iii) component view to give the reader a better understanding of the basic building blocks of this important system and the relations between these blocks.        

4. The application domain of the solution remains uncertain. Is the approach valid for any Industry 4.0 digital twin? Is this only valid for MVP? How does the design approach (e.g., microservices) support that?  

5. To facilitate common understanding of the business value and improve the research methodology, we would like to recommend adding a list of key quality attributes (e.g., scalability, usability. modifiability, security, performance etc. according to the existing software quality standards), prioritizing these attributes and relating these to the design decisions made in the case study (e.g., open-pit mining enterprise) presented.

6. The language needs improvement as a number of sentences/phrases seems too general/vague, e.g., "At the initial design stage, the first created model should be understandable to employees and, in general, can be created by themselves or with their direct participation." (p.11).

Overall, the paper would essentially benefit from a clearer narrative flow that presents the research methodology in a better understandable form and supports it by a specific case-based verification. Given its current form, the paper needs a thorough revision based on the above comments and suggestions. 

Author Response

Dear reviewer!

Thank you for your comments to improve the quality of the article! We have made a large number of edits, the answers to your comments are in the attached file.

Kind regards,

Authors!

Round 2

Reviewer 3 Report (New Reviewer)

Based on the previous comments, authors improved the paper. Minor issues still exist; however, the paper in its current form seems acceptable. In particular, this paper would essentially benefit from a brief cross-cutting use-case instantiating and supporting the approach.

This manuscript is a resubmission of an earlier submission. The following is a list of the peer review reports and author responses from that submission.

Round 1

Reviewer 1 Report

The article is word-to-word translation of two prior published articles without any new ideas or results

10.25018/0236_1493_2022_2_0_97

10.25018/0236_1493_2022_3_0_117

Author Response

Dear reviewer!

Thank you for your work and attentive attitude to the manuscripts! We are glad to know that reviewers carefully check articles for borrowing.

We will not hide that we are saddened by your decision not to give any detailed comments regarding the content and suggestions for correction. We would like to note that the Informatics policy allows to publish materials that are translations of previously published works in another language, especially since in this case the authors remain the same and this is not plagiarism of other people's materials.

Also, we would like to inform you that the presented work is not exactly a translation of the articles you indicated:

  1. The materials were subject to reconceptualization and generalization for a wider range of subject areas.
  2. Work has been done on the mistakes made earlier.
  3. Added more than 1/3 of the materials that give a better understanding of the results and link two separate works together. These materials have not been published anywhere before.

At the moment, in accordance with the Informatics policy, we have indicated in the Gratitude section information that the submitted work contains materials previously published in Russian.

In the attachments to the answer, we added a version of the article with corrections based on the comments of reviewers.

With best wishes, authors.

Reviewer 2 Report

The paper is well structured and quite informative to the readers in relation to the concepts related to enterprise architecture in the context of digital enterprises within an Industry 4.0 context. And it proposes a graphical modeling approach for da-centric microservice architecture and a so-called DEAL (prototype) tool, used in particular for the documentation of a ‘digital enterprise architecture metamodel’.

The main text itself is relatively consistent in its terminology, and relatively easy to follow; however,   there are major inconsistencies between the main text and:

1.       The title: where it refers only to ‘instruments’ (not mentioned anywhere else in the text), and ‘design’ which is by far the main topic of this paper.

2.       The abstract: where it mentions ‘unification of methods’ (unification not mentioned anywhere within the main text), reference model (not mentioned anywhere within the main text) and ‘non-functional requirements’ that is barely mentioned only once in the text and not discussed at any length.

3.       The conclusion: which instead refers to an ‘approach’ and to ‘an unified reference metamodel’.

Both the title, the abstract and the conclusion must be revisited to ensure adequate alignment of concepts and terms to verify that what has been ‘sold’ in the title and abstract have indeed been delivered by the authors in the main text.

On page 21 of the paper, the end paragraphs claim a number of advantages of the ‘proposed approach’ but ‘qualifiers’ in these claims are both vague and not really supported by evidence within the main text itself.

For advantage 1: the claim of ‘universal’ is not really supported.

For advantage 2: the claim of ‘most of the diagrams are ‘quite abstract’ and ‘easy to understand’ do not say much.

For advantage 3: ‘… and, ultimately, minimizes the risk of error occurrence’: this is stated as an ‘assertion’ while it is most probably much closer to a ‘wish’ rather than a ‘certainty’.

I understand that it is very challenging to ‘validate’ the proposals’ presented in this paper, it would have been however feasible to figure out some ‘criteria’ met by the proposals in this paper, and use such criteria in comparison with previously available ‘methods and tools’ to illustrate how the authors’ proposals meet these criteria better.

Author Response

Dear reviewer!

We sincerely thank you for your high appreciation of our work and detailed suggestions for improving the article!

We found your comments very useful and in accordance with them we carried out work to correct the materials, namely:

  1. Changed the title of the article, replacing "tools" with "method".
  2. Completely rewritten the annotation, making it more correct regarding the content of the work.
  3. We also rewrote the conclusions and corrected the terminology.
  4. We have fully taken into account your comments about the stated advantages. You are right, it is quite difficult to confirm or refute them at the current stage, so we tried to express our thoughts more correctly and with restraint, bringing them together with the initial theses about existing problems in this field of research.
  5. We have also corrected the English language in some places.

Once again, we sincerely thank you for your work!

With best wishes, authors!
